# Extracellular Vesicles (EVs) as Crucial Mediators of Cell-Cell Interaction in Asthma

**DOI:** 10.3390/ijms24054645

**Published:** 2023-02-28

**Authors:** Mariaenrica Tinè, Ylenia Padrin, Matteo Bonato, Umberto Semenzato, Erica Bazzan, Maria Conti, Marina Saetta, Graziella Turato, Simonetta Baraldo

**Affiliations:** 1Department of Cardiac, Thoracic, Vascular Sciences and Public Health, University of Padova and Padova City Hospital, 35128 Padova, Italy; 2Pulmonology Unit, Ospedale Cà Foncello, Azienda Unità Locale Socio-Sanitaria 2 Marca Trevigiana, 31100 Treviso, Italy

**Keywords:** airway inflammation, signaling, microRNA (miRNA), endotypes

## Abstract

Asthma is the most common chronic respiratory disorder worldwide and accounts for a huge health and economic burden. Its incidence is rapidly increasing but, in parallel, novel personalized approaches have emerged. Indeed, the improved knowledge of cells and molecules mediating asthma pathogenesis has led to the development of targeted therapies that significantly increased our ability to treat asthma patients, especially in severe stages of disease. In such complex scenarios, extracellular vesicles (EVs i.e., anucleated particles transporting nucleic acids, cytokines, and lipids) have gained the spotlight, being considered key sensors and mediators of the mechanisms controlling cell-to-cell interplay. We will herein first revise the existing evidence, mainly by mechanistic studies in vitro and in animal models, that EV content and release is strongly influenced by the specific triggers of asthma. Current studies indicate that EVs are released by potentially all cell subtypes in the asthmatic airways, particularly by bronchial epithelial cells (with different cargoes in the apical and basolateral side) and inflammatory cells. Such studies largely suggest a pro-inflammatory and pro-remodelling role of EVs, whereas a minority of reports indicate protective effects, particularly by mesenchymal cells. The co-existence of several confounding factors—including technical pitfalls and host and environmental confounders—is still a major challenge in human studies. Technical standardization in isolating EVs from different body fluids and careful selection of patients will provide the basis for obtaining reliable results and extend their application as effective biomarkers in asthma.

## 1. Introduction

Asthma is a chronic respiratory disorder that is estimated to affect 300 million persons worldwide, a figure that is projected to sharply increase with 100 million further cases by 2025. It is currently the most common chronic condition in the pediatric population, affecting 5–10% of children and adolescents [1]. Although some studies suggest that the rise may have subsided in some countries, asthma continues to pose a substantial socio-economic burden (impact on quality of life, days lived with disability, absence from school and from work). The increase in asthma prevalence has been paralleled by a similar increase in other allergic conditions such allergic rhinitis and eczema. 

Clinically, asthma is defined by a history of respiratory symptoms that include wheezing, shortness of breath, chest tightness, and cough, and is associated with variable airflow limitation. Both symptoms and airflow limitation characteristic of asthma vary over time and in intensity [2] and can be triggered by several factors including allergen exposure, exercise, laughter, irritants, weather changes, and viral infections [2]. The delineation of comorbidities is also an important share in the burden of the disease, because comorbidities can contribute to poor control of asthma and may affect the natural history of the disease and the therapeutic approach [3,4]. Major impact comorbidities are: (1) rhinitis (particularly allergic rhinitis), which results in upper airway inflammation and contribute to exacerbate asthma [5]; (2) gastroesophageal reflux which may aggravate airway inflammation [6]; (3) obesity, which affects respiratory mechanics but also induces a systemic pro-inflammatory status [7]; and (4) bronchiectasis, which increases the risk of exacerbations due to pulmonary infections [8].

Asthma is a heterogeneous disease with different underlying pathogenetic pathways [2] that results from complex interactions between individual genetic susceptibility, host factors, and environmental exposures. Defining these pathways, the asthma endotypes, has become a key task in asthma, as the endotype-driven approach offers a way to better diagnose, monitor, and refer patients to the most appropriate therapeutic strategies [9]. Airway inflammation is the main recognized mechanism, and the typical inflammatory setting is characterized by the presence of eosinophils in the submucosa. Other cell types that populate the inflammatory infiltrate are mast cells, basophils, neutrophils, monocytes, and macrophages [10]. The canonical inflammatory response—the so-called type 2 inflammation—is fueled by T helper 2 lymphocytes that produce Interleukin (IL)-5, IL-4, and IL-13. It occurs in as many as 80% of children and it is also seen in the majority of adults with asthma in association with sensitization to environmental allergens, such as those from dust mites, fungi, pets, and pollens [11,12,13,14]. Besides classic allergic asthma, T2-high asthma also includes aspirin-exacerbated asthma with chronic rhinosinusitis/nasal polyposis. 

On the other side of the spectrum, T2-low asthma (sometimes referred to as non-eosinophilic asthma) encompasses both inflammatory endotypes where T2 cytokines are not involved in driving asthma pathobiology: neutrophilic and paucigranulocytic. Neutrophilic asthma is usually associated with obesity and the key cytokines involved are IL-17, IL-8, and IL-6 [15,16].

Paucigranulocytic asthma is characterized by the presence of significant airway remodeling in the absence of concomitant airway inflammation. Bronchial wall remodeling is another key pathogenetic trait of asthma that has been described as an abnormal tissue reparation process responsible for chronic lesions to the normal airway structure. It is now evident that remodeling occurs very early in the natural history of asthma, starting in infancy before the age of two years [17,18] and may have a negative impact on long-term functional outcomes. The most important remodeling features include epithelial shedding, fibrosis of subepithelial basement membrane (BM thickening), smooth muscle hypertrophy/hyperplasia, and neoangiogenesis [19,20] (Figure 1).

In the classic T2-high asthma, the epithelial-derived cytokines initiate the process by driving dendritic cell activation and phenotypic changes in the airways, followed by their migration to secondary lymphoid tissues where they present the allergen to naïve T cells and orient them towards a T2 profile, characterized by the production of IL-4, IL-5, and IL-13 among others.

The main target of IL-5 is the eosinophil: through its receptor, IL-5 can activate different signal transduction pathways, inducing growth, bone marrow maturation, peripheral migration, and the activation and survival of eosinophils [21]. On the other hand, IL-4 is mainly directed toward B lymphocytes with activation, immunoglobulin production, and isotype switching towards IgE. IL-4 also induces the differentiation of CD4^+^ T lymphocytes into the Th-2 subtype with an autocrine mechanism and acts on structural cells (epithelium, goblet cells, fibroblasts, smooth muscle) [22,23]. IL-13 has an action partially redundant to IL-4: indeed, they share a common receptor (IL4/IL-13R) and the same targets (mainly B lymphocytes, epithelium, endothelium, smooth muscle). IL-4/IL-13 are therefore implicated both in the amplification of the inflammatory process and have a strong involvement in the processes of airway remodeling and neoangiogenesis [24].

Innate immune responses were traditionally considered responsible for the release of acute mediators of inflammation during an asthma attack. Resident innate cells (mainly mast cells and macrophages) produce histamine, serotonin, and substance P that are stored in preformed granules and immediately released after the encounter with the triggering factor. They will promote the contraction of smooth muscle, production of mucus, and increase in vascular permeability, which results in edema [25].

More recently, innate immune responses have gained a central stage in asthma also for their immune modulatory role after the discovery of type 2 innate lymphoid cells (ILC2) [26,27,28]. Unlike lymphocytes, ILCs do not express antigen-specific receptors, but are instead activated by a broad range of signaling molecules. They concentrate at barrier surfaces such as the skin, gut, and airways, at which site ILC2s are the dominant subset [29,30,31,32]. Upon exposure to cytokines, allergens or viruses, ILC2s accumulate in the submucosa, close to epithelial cells and T cells, and have been involved in the development and maintenance of type 2 inflammation in asthma [33,34,35,36,37]. In the context of innate responses, alarmins, e.g., IL-25, Thymic stromal lymphopoietin (TSLP) and IL-33, are the most important players.

IL-25 is expressed as a preformed cytokine, stored in granules, by the airway epithelium and is rapidly released upon cell stimulation by environmental triggers, including allergens [38]. TSLP is also released in response to epithelial stimuli (e.g., allergens, viruses, bacteria, pollutants, and smoke) and initiates multiple downstream innate and adaptive pathways involved in asthma. Inhibition of TSLP represents a novel approach to treat the diverse endotypes of asthma [38].

IL-33 is a central component for activation of both the innate and adaptive arms of immunity [39]: it is responsible for inducing early immune development and polarization toward type 2 T cell inflammation [40] through activation of resident dendritic cells (DC), but also independently of DCs [41].

Of interest, there is recent evidence suggesting that IL-33 is essential to the process of airway remodeling observed in asthma through the upregulation of cluster of differentiation (CD)146, thus promoting epithelial mesenchymal transition [42]. The contribution of epithelial tissues to fibrosis through the process of epithelial–mesenchymal transition (EMT) has previously been demonstrated in lung cancer as well as in fibrotic diseases (of the lung, kidney, eye, liver, and intestine). These data expanded the knowledge of the mechanisms regulating the EMT process in asthma and substantiate the hypothesis that structural abnormalities in the asthmatic airway epithelium could lead to enhanced signaling to underlying mesenchymal and immune cells, driving the abnormal responses to environmental stimuli in the asthmatic airway [43].

## 2. Extracellular Vesicles

In such a complex pathogenetic scenario, where several cell types cooperate to provoke the clinical manifestations of asthma, understanding the mechanisms of cell–cell interactions rises up as a crucial issue. In parallel with canonical signaling, a new army of submicroscopic messengers has been identified and is catalyzing scientific interest: The extracellular vesicles (EVs). Chargaff et al., in 1946, first argued the existence of cytoplasmic debris released from platelets characterized by functional thromboplastic activities [44]. Later, in 1967, the advent of electron microscopy allowed for the identification of such components, minute platelet-derived particulates rich in lipid content they termed “platelet dust” [45]. In the following decades, small vesicles secreted or shed by the cell membrane were detected in cell culture, plasma, and body fluids. According to the size, origin, and surface markers [46], these particles were distinguished into:exosomes: 30–200 nm diameter; generated by inward budding of the membrane (endocytosis), subsequent formation of multivesicular bodies, and release by exocytosis; and characterized by tetraspanins (CD9, CD63, CD81, and CD82) and other surface markers derived from the multivesicular body.microvesicles: 100–1000 nm diameter; released by budding and shedding from the plasma membrane of activated cells. They share the same membrane components with the parent cell.apoptotic bodies: 1000–4000 nm diameter; released through blebbing of apoptotic cell membrane by cells undergoing apoptosis; and enriched in phosphatidylserine, annexin V.


These different types of vesicles are exemplified in Figure 2 in a bronchial epithelial cell.

Nonetheless, methodological issues (i.e., lack of specific markers, overlapping vesicles size) and the trend to focus on their content more than on their classification supported the recommendation of using of the umbrella term “extracellular vesicle” to describe all the anucleated particles naturally released from the cell delimited by a lipid bilayer [47,48]. Indeed, the charm of EVs stands in their ability to mediate intercellular cross-talk and quietly influence the inflammatory milieu. Once released, the EVs can be internalized via endocytosis or membrane fusion, releasing their contents into “recipient” cells [49]. EV cargo includes proteins, sugars, lipids, immune molecules (MHC), cytokines, hormones, and a wide variety of genetic materials, such as DNA, mRNA, and non-coding RNAs, a content protected from proteases and nucleases in the extracellular space by the limiting membrane [50]. EVs are able to convey cell messages not only to the tissue microenvironment but also throughout the body, representing both harmful and homeostatic ambassadors.

Circulating blood cell-derived EVs were initially linked to coagulopathies and cardiovascular disorders [51]. A peculiar increase in circulating EVs is common among different conditions associated with increased cardiovascular risk factors, such as smoking [52], diabetes mellitus [53], and hypertension [54]. Indeed, EVs might exert a pro-inflammatory action favoring atherosclerosis development, as suggested by some in vitro evidence. EVs can stimulate cytokine release from endothelial cells promoting monocyte adhesion to the endothelium and their migration to the plaque [55,56] and increase endothelial permeability [57]. These mechanistic in vitro studies help with understanding the importance of these vesicles in the development of atherosclerotic lesions. In fact, both microparticles [58] and exosomes [59] have been described in the plaque, and the levels of circulating procoagulant EVs are increased in patients with acute coronary syndrome compared to those with stable coronary disease and healthy controls [60,61,62]. Increased levels of pro-coagulant microparticles—filled with tissue factor and phosphatidylserine—have long been detected in the plasma of patients with pulmonary embolism [63,64,65,66]. Recently, the abundance of platelet and leukocyte-derived EVs was acknowledged as a possible contributor in hypercoagulability and trombophilic risk in severely ill COVID-19 patients [67]. Beyond coagulopathies, an increase in peripheral EV level was observed in many inflammatory conditions, including connective tissue diseases [68], inflammatory bowel diseases [69], and vasculitis [70].

There is accumulating evidence that EVs may be key players in the respiratory system as well, in regulating both homeostatic and pathologic conditions [71], especially in lung cancer where EVs represent not only a perfect target for liquid biopsy but also key effectors of tumor development as key regulators of tumor microenvironment [72].

We will then review the current knowledge delineating the possible contribution of EVs in the pathogenetic mechanisms of asthma: From the evidence in preclinical studies to their detection in tissue, blood, Bronchoalveolar Lavage (BAL), and other samples of patients with asthma with the intent to explore their possible applications from identification of specific disease endotypes to monitor disease activity and evaluate response to therapies.

## 3. Mechanistic Hints from Preclinical Studies

Accumulating evidence from preclinical studies supports the hypothesis that EVs are crucial mediators of cell–cell interaction in asthma as in many other immune-mediated diseases [73]. In most studies, EVs produced by the immune cells were the main focus of research in the EV field. However, multiple kinds of non-immune cells have been shown as efficient EV sources: epithelial cells, fibroblasts, smooth muscle cells, and mesenchymal cells. These are often significant contributors to the ongoing immune response.

### 3.1. Epithelial Cells

Pulmonary epithelial cells appear to be the main source of EVs in the lungs [74] (Figure 2). Such production is not steady since lung inflammation can significantly alter EVs content and release from epithelial cells, as observed in cell cultures [75]. Different well-known triggers of inflammation in asthma have the potential to modify EV release from airway epithelial cells:

**Allergic inflammation**: Increased levels of EVs, mainly of epithelial origin, were detected in the BAL of mice with allergic airway inflammation compared with controls [76]. Such a finding was later confirmed by complementary techniques in another study that analyzed the EV content of BAL fluid in mice. The authors observed that 80% of the alveolar EVs were of epithelial origin and that, after lung allergen challenge, hematopoietic cell-derived EVs doubled along with miRNAs selectively expressed by immune cells, such as miR-223 and miR-142a [77], supporting the key interplay between immune and non-immune cells mediated by lung EVs. Indeed, when treated with IL-13, bronchial epithelial cells released exosomes that promoted monocyte proliferation and chemotaxis [76]. A subsequent study suggested that epithelial-derived EVs released under IL-13 stimulation are enriched in 16 specific miRNAs that might influence Th2 polarization and dendritic cell maturation [78].

**Mechanical stress**: Bronchoconstriction, simulated in vitro by a compressive stress under regulated pressure, triggers the release of EVs bearing Tissue Factor, a molecule specifically increased in the BAL of asthmatics compared to healthy controls [79]. Mechanical compression of human bronchial epithelial cells is associated also with release of tenascin C carrying EVs [80]. Tenascin C is an extracellular matrix glycoprotein that modulates cellular processes such as cell adhesion, proliferation, and migration during embryonic development and tissue repair. Its expression is mostly absent in physiologic conditions and increases with tissue remodeling and disease, particularly in asthma, where tenascin C is abundant in the subepithelial basement membrane at baseline and further increased in response to allergen challenge [81].

**Respiratory infections**: Respiratory tract infections are common causes of asthma exacerbations. Rhinovirus infection can promote the release of EVs containing tenascin-C, which, in addition to the already enlisted functions, activates local cytokine synthesis. Such pathways are highly activated in the airways of people with asthma upon virus stimulation [82]. Furthermore, exosomes isolated from cells infected with respiratory syncytial virus are able to activate innate immune responses by inducing cytokine and chemokine release from human monocytes and airway epithelial cells [83]. Animal models suggest that EVs released during infectious events have mainly a pro-inflammatory role and favor persistent lung damage. Indeed, a first study in 2010 suggested that, after lipopolysaccharide (LPS) stimulation, epithelial-derived EVs are enriched in BAL and that these EVs are characterized by surface exposure of the inflammatory marker, Major Histocompatibility Complex (MHC) class II [84]. Conversely, it was later observed that sterile stimuli (oxidative stress, acid aspirations) mainly induced the accumulation of epithelial-derived-EVs in mice BAL [85], whereas infectious stimuli (LPS, gram negative bacteria) mainly promoted the release of alveolar macrophage-derived EVs [86]. Whatever the prevailing parental cell, EVs released during an infectious event are known to activate and amplify the inflammatory response. Indeed, LPS-induced EVs were found to enhance the production of Th1- and Th17-polarizing cytokines (IL-12p70 and IL-6, respectively) by lung dendritic cells [84]. On this line, IL-17A and Tumor Necrosis (TNF) alpha co-stimulation of epithelial cells was found to promote the accumulation of EVs able to enhance neutrophil chemotaxis [87].

In addition, the EVs released in the lung under infectious conditions might not only be amplifiers of infective exacerbations but might also be implied in infection prevention/resolution. Indeed, Kesimer et al. observed that epithelial cells released exosome-like vesicles with a neutralizing effect on human influenza virus [88]. Moreover, epithelial cell-derived EVs contain membrane mucins that are known to be part of the mucociliary clearance systems and innate immunity that protects the respiratory tract from environmental pathogens and xenobiotics [89].

**Pollutants**: air pollution has been proven, both in vitro and in vivo, to elicit EVs release [90]. Nonetheless, the quality of available evidence is limited by the wide variety of models, time of exposure, and EV characterization techniques. Stassen et al. have shown that EV production is induced regardless of Particulate Matter (PM) size (PM2.5 or PM10) but the mechanism involved seems different and could imply discrepancies in the ability of these PMs to form reactive oxygen species (ROS) [91]. Neri et al. reported that a large amount of EVs are secreted by mononuclear and endothelial cells treated with urban PM [92]. Furthermore, pollutants exposure is known to have a strong impact on vesicular miRNA content both in vitro and in vivo [93,94].

**Cigarette smoke**: A number of studies have shown that the exposure to cigarette smoke extract leads to increased secretion of EVs from cultured human bronchial epithelial cells [95,96]. Such EVs can mediate interaction with the airway vasculature by promoting endothelial cell survival [97]. These and other studies (summarized in reference [98]) suggest that EVs induced in response to cigarette smoke might modulate inflammation, thrombosis, tissue remodeling, endothelial dysfunction, and angiogenesis.

As described above, several triggers can promote EV release from epithelial cells. Further insights on the functional significance of epithelial cell production of EVs are suggested by a recent study which compared the differential EV release from the apical and basolateral side of bronchial epithelial cells from either asthmatics or healthy controls. In an air–liquid interface model, the authors isolated EVs from both the apical and the basolateral side, finding significant differences. Of interest, the EV populations isolated from the apical cell side were mainly composed of vesicles with diameters matching the size range of the exosomes. Conversely, on the basolateral cell side, median vesicle diameters were noticeably larger, consistent with the size of the microvesicles rather than with the exosomes. Furthermore, 236 miRNAs were differentially expressed depending on the EV secretion side, regardless of the disease phenotype. A pathway analysis predicted mTOR (mammalian target of rapamycin) and MAPK (mitogen-activated protein kinase) signaling pathways as potential downstream targets of apically secreted miRNAs. In contrast, miRNAs specifically detected at the basolateral side were associated with pathways of T and B cell receptor signaling, suggesting a profound effect on the regulation of submucosal inflammation. Even if differences in miRNA profiles were more pronounced in comparing apical versus basolateral sides, some alterations were also observed in the comparison of asthmatics with healthy controls in the levels of certain EV-associated miRNAs (or their families). The study proves a compartmentalized packaging of EVs by bronchial epithelial cells supposedly associated with site-specific functions of cargo miRNAs, which are considerably affected by disease conditions such as asthma [99].

### 3.2. Immune Cells

In vitro studies have shown that EVs are released by all the main inflammatory cell types involved in asthma pathogenesis including mast cells [100], eosinophils [101,102,103], monocyte/macrophages [104], neutrophils [105], T lymphocytes [106], and dendritic cells [107,108].

**Eosinophils** purified from peripheral blood have the potential, when stimulated with Interferon (IFN)γ, to release EVs [109]. Eosinophil-derived EVs were shown to up-regulate reactive-oxygen species and nitric oxide production in eosinophils and to enhance their chemotaxis and adhesion by Intercellular Adhesion Molecule 1 (ICAM-1) and integrin-2 [102]. Along this line, the same group observed that eosinophil-derived EVs could increase the apoptosis of small airway epithelial cells and reduce their wound healing capacity [103].

Similarly, **mast cells**, which naturally store histamine, proteoglycans, and other fast acting inflammatory mediators [110], were shown to release EVs in blood capable of boosting lung allergic reaction [111].

As for **alveolar macrophages**, it was shown that mouse resident AM could blunt inflammatory signalling in alveolar epithelial cells by transcellular delivery of suppressor of cytokine signalling 3 (SOCS3) within EVs [112]. Indeed, SOCS3 levels were reduced in a murine model of chronic asthma and macrophage-derived EVs enriched in SOCS3 inhibited the activation of transcription in epithelial cells challenged with IL-4/IL-13 [113]. Primary human macrophages and dendritic cells have the potential to promote granulocyte migration through EV release [104]. They are also active on T-lymphocytes by the release of exosomes expressing OX40 ligand (OX40L), which is able to promote proliferation and differentiation of CD4^+^ T cells towards a Th2 phenotype [114].

Supporting further implications of involvement of EVs in asthma, exosomes with the potential to be pathogenic were isolated from lymphocytes as well. Indeed, EVs isolated from **B cells** can present allergen-derived peptides to T cells and induce their proliferation and production of Th2-like cytokines [115] either in a direct manner [116,117] or in cooperation with dendritic cells [118,119]. Furthermore, **T cells** could promote mast cell degranulation even at distant inflammatory sites by releasing EVs in the bloodstream [120].

In contrast with eosinophilic asthma, which is extensively investigated, **neutrophilic** inflammation characterizes the most challenging phenotype of asthmatic patients [121,122]. The studies aiming at investigating EVs released from neutrophils are scarce in asthma. Nevertheless, emerging evidence suggests that neutrophil-derived EV might promote the growth of airway smooth muscle cells, thus favoring airway remodeling, as suggested by co-cultures of neutrophil-derived exosomes and smooth muscle cells in a natural model of equine asthma [105,123]. 

### 3.3. Other Non-Immune Cells

At variance with the growing evidence on inflammatory/immune cell-derived EVs, less is known on the role of the EVs released from non-immune cells in asthma development.

The EVs detected in the culture media from severe asthmatics’ **fibroblasts**, mainly exosomes, have been shown to be filled with cytokines and chemokines that are easily uptaken by bronchial epithelial cells promoting their proliferation. Of interest, the mRNA and protein levels of transforming growth factor (TGF)-β2 secreted from fibroblasts were similar in patients with asthma compared to controls, whereas TGF-β2 content in exosomes was differentially expressed [124]. Such discrepancy suggests that exosomal carrying of TGF-β2, a well-known regulator of cellular growth, is specifically impaired in severe asthma and could contribute to airway remodeling.

Indeed, available asthma treatments can mostly control inflammation but have limited effects on airway remodeling, which cannot be reversed [125]. Promising results addressing airway remodeling come from studies in mice in which **mesenchymal cells** were able to reduce smooth muscle hypertrophy and vascular hyperemia [126]. Recent evidence suggests that the systemic administration of EV derived from mesenchymal cells could vehicle beneficial effects in asthma (as do the intact parental cells), including reduced collagen fiber deposition and decreased TGF-β levels in lung tissue [127]. When co-cultured, adipose tissue mesenchymal stromal cells release exosomes that can be internalized by airway smooth muscle cells, promote smooth muscle cells apoptosis and suppress inflammatory mediators secretion from muscle cells [128]. Dong et al. observed that the administration of mesenchymal cell-derived EVs, especially when released under hypoxia, prevented mouse chronic allergic airway remodeling, as suggested by the decreased expression of pro-fibrogenic factors α-smooth muscle actin, collagen-1, and the TGF-β1-p-smad2/3 signaling pathway [129]. Collectively, these results support the prospect that mesenchymal cell-derived EVs will provide a future therapeutic approach in respiratory diseases, as reviewed in reference [130].

An overview of current evidence on EVs functions from immune and non-immune cells is summarized in Table 1.

## 4. Extracellular Vesicles as Asthma Sensors in Humans

Today, the search for effective biomarkers, meaning easily measurable and reproducible sensors of disease phenotype and prognosis to personalize the approach to asthma patients, has fostered clinical and scientific interest in EVs. Indeed, the presence of these particles in body fluids has long been related to specific disease, suggesting their use as sensors of disease activity. The first studies on asthma EVs were inspired from those on cardiovascular diseases [63,64,65,66] and focused on circulating microparticles. Such particles share surface markers with parental cells, a feature that makes them easily detectable in blood by immunoassays and cytometric analysis. Duarte et al. compared circulating levels of platelet and endothelial-derived microparticles in asthmatics and control patients and found that the number of platelet-derived EVs was higher in asthma [131]. On the same line, high levels of circulating endothelial-derived microparticles (CD31^+^/CD41^−^) and other inflammatory markers have been reported in asthmatic patients in Beijing and are associated with air pollution. Following the remarkable improvement in air quality achieved during the 2008 Olympic games, microparticles decreased in asthmatics and reached controls’ levels, a consistent benefit that was confirmed after 2 months [132]. Nonetheless, a similar involvement was observed in several inflammatory diseases [68,69,70,133,134] mining the specificity of blood-derived microparticles as a biomarker specific for asthma. Wagner et al. characterized plasma EVs according to surface markers, particles and protein concentrations, and cytokine content in allergic patients. By this approach, they showed that the pro-allergic cytokines IL-4 and IL-5 were higher in plasma EVs of patients with allergies than in healthy controls combined with the pro-inflammatory cytokines IL-6 and TNFalpha [135]. Another valuable option is the search for vesicular miRNA profiles. Plasma exosomal miR-124, miR-125b, miR-133b, miR-130a, and miR-125b-1-3p differed between asthmatics and controls and were related to CRP and IgE levels [136]. Of particular interest was miR-125b, who’s levels not only differed between patients and controls but also significantly increased with asthma severity (intermittent, mildly persistent, moderately persistent, and severely persistent asthma) [137]. However, another study detected no differences in miR-125 levels in moderate asthmatic patients compared to the healthy controls, whereas miR-223 and miR-21 were significantly up-regulated [138]. Plasma EV miR-122-5 was increased in patients with uncontrolled asthma compared to controls, and its expression was related to eosinophil and neutrophil count [139]. Of note, the levels of mi-RNA-126, which are increased in peripheral blood exosomes of patients with atopic asthma, were also increased in the lungs of an allergic asthma mouse model [140]. Another interesting finding is that inflamma-miRNAs (i.e., miRNA released by inflammatory T cells) could label distinct endotypes of asthma, being differentially expressed in serum exosome samples of T2 high and T2 low asthmatic patients [141].

Although these approaches represent a significant step forward supporting the application of circulating EVs as biomarkers of disease endotypes in asthma, peripheral blood is still far from being a reliable surrogate of lung pathology and inflammation [17,18]. Limited but valuable evidence is available on the presence and the main immunological role of EVs in the BAL of asthmatic patients. Admyre’s group firstly isolated EVs with major histocompatibility complex class II in healthy volunteers’ BAL, suggesting that floating EVs might activate the T-cell mediated immune response [142]. Hough and colleagues showed that asthmatics’ EVs presented increased frequencies of Human Leukocyte Antigen (HLA)-DR, an MHC-II molecule as compared to healthy controls, and that EVs concentrations were related to eosinophilia and serum IgE [143]. Furthermore, the EVs isolated in asthmatics’ BAL were characterized by lipid abundance, suggesting that they might actively transfer lipidic inflammatory mediators to alveolar cells [143]. On this line, BAL exosomes from asthma patients contain enzymes for leukotriene biosynthesis and have been proven, in vitro, to promote leukotriene 4 and IL-8 release from bronchial epithelial cells [144]. Moreover, Levanen and colleagues identified a pool of 24 exosomal miRNAs isolated from BAL that differed between asthmatics and control subjects [145].

Bronchoalveolar lavage, though informative, has a limited use both for research and in routine clinical practice, especially in severe asthma. Looking for less invasive techniques, Bahmer and colleagues profiled the EV-miRNA signature in plasma of patients with asthma, reporting differential expression of several miRNAs, particularly miR-122-5p. In a pilot study performed in sputum, they confirmed the results obtained in plasma EVs [139]. A recent study applied proteomic analysis to analyze the aerosol of droplet particles that are formed from the epithelial lining fluid when the small airways close and re-open during inhalation succeeding a full expiration detected. The asthmatic patients’ proteome, compared to healthy controls, was enriched in extracellular proteins associated with extracellular exosome-vesicles and innate immunity [146]. Furthermore, a reduced expression of miR-34a, miR-92b, and miR-210 was detected in the nasal lavage of asthmatic children compared to healthy controls, and their levels were associated with the obstruction of large and small airways [78]. In addition to sputum, droplets, and nasal lavage, the presence of EV was recently confirmed also in the saliva of a cohort of children with asthma, opening further opportunities for research in this emerging landscape [147].

## 5. EV in Our Guests: How Microbiome Introduces Higher Levels of Complexity

Host’s EVs are not alone in the lung milieu. Of interest, airway and gastro-intestinal tract microbiota show peculiar aspects in asthma, and accumulating evidence supports the existence of a link microbiome—inflammatory endotypes as well as a possible role of microbiome in disease pathogenesis [148,149]. The following studies have tested the potential of EVs to mirror the complexity of the human microbiota. By 16 s rDNA amplification and sequencing of plasma EVs isolated from 260 patients with asthma and 190 healthy controls, Lee et al. caught a frame of the bacterial composition. Whether the increased abundance of bacteroidetes in asthmatics’ plasma results from differences in the microbiota of the lung or gut is still unknown since no parallel assessment of respiratory/intestinal samples was performed [150]. Overcoming the intrinsic limitations of studying the microbiome on plasma EVs, An et al. focused on the exhaled breath condensate obtained from 58 healthy controls and 251 patients with asthma. Based on EV populations, they found higher complexity and biodiversity in asthma but no association was identified between airway microbiota and specific inflammatory/clinical phenotypes [151]. Further expanding such investigations, metagenomic analysis of EV has been applied to the urine samples collected from children with allergic airway disease who shared unique features when compared to atopic and healthy controls [152,153]. Children with allergic airway disease shared the highest level of urine EVs derived from Klebsiella and Haemophilus, and their level was positively related to total IgE and eosinophil percentage [153].

Nucleic acid sequencing of EV cargo detected in non-invasive samples might offer, in the next feature, a novel platform to study dysbiosis in the pathogenesis of asthma. Nonetheless, studies are still in their infancy, and several concomitant confounding factors need to be ascertained when investigating bacteria-derived EVs, including environmental and dietary factors.

## 6. Conclusions

Accumulating evidence on EV features, biogenesis, and release in asthma hold out promises for effectively understanding the role of these subcellular particles in disease progression. EVs are released in the lung microenvironment by both immune and structural cells, mainly epithelial cells and fibroblasts. In the context of epithelial responses, the packaging of EVs is associated with the distinct functions of their cargo miRNAs depending on the specific site of release (damage sensing in the apical side, immune regulation in the basal side). EVs have been proven in experimental models to modulate several pathways, mainly through the delivery of miRNAs, proteins, and lipid mediators. Collectively, available evidence mainly suggests a pro-inflammatory and pro-remodelling role of EVs in asthma pathogenesis, whereas a minority of reports indicate protective effects. Of particular interest are mesenchymal cells-derived EVs, which have shown promising results when administered as a therapeutic strategy to contrast airway remodeling.

Unraveling the role of EVs in different clinical phenotypes of asthma will allow to exploit their high diagnostic and prognostic potential as biomarkers. In an attempt to improve their specificity as biomarkers, researchers are working on identifying unique combinations of surface markers and cargoes that characterize distinct EVs subtypes. Furthermore, biological samples reflecting more directly the airway milieu rather than peripheral blood are emerging as a source for the study of EVs: bronchoalveolar and nasal lavage, sputum, saliva, and droplets from the epithelial lining fluid. Expanding knowledge indicates that EV release can be potentially influenced by a vast number of factors, all relevant to the pathogenesis of asthma, including repeated mechanical stress, respiratory infections, air pollution, cigarette smoke, and microbiome. Thus, an exhaustive phenotypical characterization of the study subjects and a detailed evaluation of all possible confounding factors are essential to decipher the language of EVs in asthma.

## Figures and Tables

**Figure 1 ijms-24-04645-f001:**
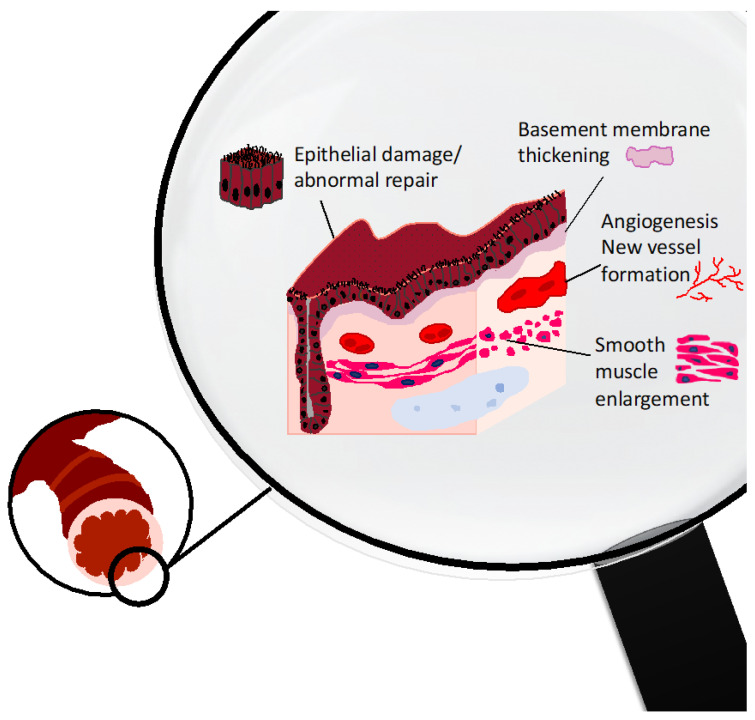
The main pathological hallmarks of remodeling in asthma encompass alterations in different airway compartments: Airway epithelium (epithelial shedding, mucous cell metaplasia); subepithelial basement membrane (collagen deposition and thickening of the reticular layer); blood vessels (neoangiogenesis); and airway smooth muscle (enlargement due to hypertrophy/hyperplasia).

**Figure 2 ijms-24-04645-f002:**
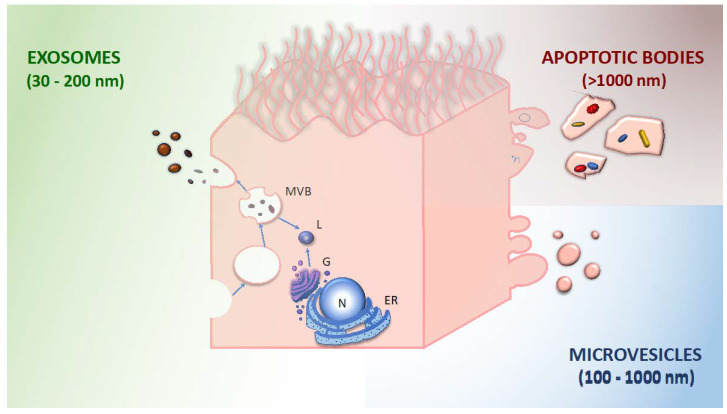
Epithelial cells have the potential to release extracellular vesicles with different sizes, origins, and cargoes. Within the endosomal system, internalized cargoes are sorted into early endosomes that then mature into multivesicular bodies (MVB). Cargoes are also delivered from the trans-Golgi network (G) and from the cytosol. Multivesicular bodies cargoes get transported to the plasma membrane, fuse with the cell surface, and are secreted in the extracellular space as exosomes. At variance, microvesicles refer to vesicles generated via direct outward blebbing and pinching of the plasma membrane, a phenomenon accompanied by distinct, localized changes in plasma membrane protein and lipid components that will result in alteration of the membrane curvature and rigidity. Finally, cells that undergo apoptosis go through membrane blebbing and disintegration of the cellular content, which will produce apoptotic bodies. ER: endoplasmic reticulum; N: nucleus; L: lysosome.

**Table 1 ijms-24-04645-t001:** Immune and non-immune cell-derived EVs and their functions in the context of asthma.

Parental Cell	Evidences on EV Release and Function	References
*Immune cells*		
Eosinophils	Eosinophil-derived EVs in asthma promote epithelial apoptosis, muscle proliferation, and eosinophil chemotaxis. IFN-γ induced eosinophil EVs are increased in asthma.	[102,103,109]
Mast cells	EVs released by FcεRI-engaged Mast Cells contain FcεRI/IgE/antigen complexes and amplify allergic responses by allowing antigen persistency. Mast cell-derived EVs, enriched in MHC class II, ICAM-1, and CD86, induce B and T cell proliferation.	[100,111]
Alveolar macrophages	Macrophage-derived EVs inhibited the activation of STATs in epithelial cells through SOCS. Macrophages-derived EVs promote granulocyte migration.	[104,112]
B lymphocytes	B-cells release EVs enriched in MHC class II molecule that induce antigen-specific T cell response.	[117]
T lymphocytes	T cell-derived EVs favour mast cell activation carrying miR-4443.	[106,120]
Neutrophils	EVs released by LPS-activated neutrophils enhance airway smooth muscle proliferation through immune-related proteins, including tenascin-X.	[105,123]
Dendritic cells	EVs secreted by dendritic cells (DC) induce antigen-specific CD8^+^ T-cells and antigen-specific IgG production. EVs released by DC promote CD4+ T cell proliferation and Th2 differentiation through OX40L.	[108,114]
*Non immune cells*		
Epithelial Cells	Pulmonary epithelial cells are the main source of EVs in normal lungs, further amplified by inflammatory stimuli. In Th2 setting epithelial-derived EVs, enriched in specific miRNAs, stimulate airway inflammation and remodeling. EV populations isolated from the apical side differ from the basolateral (in size and miRNA content).	[74,75,78,81,99]
Fibroblasts	Fibroblasts-derived EVs from asthmatics expressing lower levels of TGF-β2 increase epithelial cell proliferation.	[124]
Mesenchymal Cells	Adipose-derived EVs, which contain miR-301a-3p, limit smooth muscle cells proliferation and migration. Hypoxic umbilical cord mesenchymal cell-derived EVs, enriched in miR-146a-5p, attenuate allergic airway inflammation and remodeling in a chronic asthma model.	[128,129]

EVs: extracellular vesicles; FcεRI: high-affinity IgE receptors; STAT: signal transducer and activator of transcription; SOCS: suppressor of cytokinesignalling; BAL: bronchoalveolar lavage; LPS: lipopolysaccharide.

## Data Availability

Not applicable.

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
