# Peer review of "Extracellular Vesicles (EVs) as Crucial Mediators of Cell-Cell Interaction in Asthma"

_ijms, 2023, doi:10.3390/ijms24054645_

Round 1

Reviewer 1 Report

In this manuscript, Mariaenrica Tinè et al. describe in a very elegant way by a systemic review the participation of VEs in the microenvironment (cell-cell interaction), they begin by explaining in a general way the pathogenesis of asthma, followed by the participation of EVs derived from different cells.

2. Do you consider the topic original or relevant in the field? Does it address a specific gap in the field?

Yes, they are few reviews on the topic, and it’s an emerging area in the asthma disease

3. What does it add to the subject area compared with other published material?

They describe the participation of VEs derived from epithelial, eosinophils, B, etc cells, in this orchestration of dysregulation and how different stimuli lead to asthma.

4. What specific improvements should the authors consider regarding the methodology? What further controls should be considered?

1 Several abbreviations are not defined at the first occurrence

2 Some parts of the text are in bold line 419, and with a larger font number line 429 and 433

3 In the title you have Evs, please homogenize them with EVs.

5. Are the conclusions consistent with the evidence and arguments presented and do they address the main question posed?

Authors should Improve the conclusion and abstract with respect to the most important findings of the review.

6. Are the references appropriate?

Yes, they are.

7. Please include any additional comments on the tables and figures.

The images at first glance seem to be basic and do not provide much information, the suggestion would be, in fig 1, to detail more the pathogenesis in the image without reading the image footnote.

fig 2. Add the size, markers, etc. of the EVs.

It would also be helpful to add a table showing the cells (epithelial, eosinophils, B, T, FB...) with the most abundant proteins or miRNA or those whose function has been proven and what the function of the EVs.

Author Response

Response to reviewer 1

We thank reviewer 1 for his/her appreciation of our manuscript. The response to the specific points raised are detailed below:

Reviewer 1 Point1: Several abbreviations are not defined at the first occurrence

Response: All abbreviations are now defined at first occurrence.

Reviewer 1 Point2: Some parts of the text are in bold line 419, and with a larger font number line 429 and 433

Response: thanks for picking up this inconsistency, we have edited the font size to a uniform one.

Reviewer 1 Point3: In the title you have Evs, please homogenize them with EVs.

Response: thanks, we have edited the spelling of EVs in the title.

Reviewer 1 Point4: Authors should Improve the conclusion and abstract with respect to the most important findings of the review.

Response: we have redrafted the conclusions paragraph in the abstract and in the main manuscript in order to better reflect the most important findings of the review.

Reviewer 1 Point5: The images at first glance seem to be basic and do not provide much information, the suggestion would be, in fig 1, to detail more the pathogenesis in the image without reading the image footnote.

Response: we thank the reviewer for this suggestion. We now edited figure 1, summarizing the main structural alterations directly in the image, which is now self-explaining.

Reviewer 1 Point6: fig 2. Add the size, markers, etc. of the EVs.

Response: we have added the range of size of EVs to figure 2. Since there is no universal agreement  on the specific markers, we have chosen not to include them in the figure.  

Reviewer 1 Point7: It would also be helpful to add a table showing the cells (epithelial, eosinophils, B, T, FB...) with the most abundant proteins or miRNA or those whose function has been proven and what the function of the EVs.

Response: we have added a Table (Table 1) summarizing the main functions of EVs isolated from different cell types in the context of asthma.  

Reviewer 2 Report

In the Manuscript by Tinè et al., the authors revised literature data on the role of extracellular vesicles (EVs) in the cellular communication in Asthma. Authors introduced asthma features, EVs characteristics and classification, then they illustrated the reported features of EVs released from the different cell types involved in the asthma development (i.e., epithelial, immune and non-immune cells). Then, authors revised the data on EVs as asthma sensor by reporting the results on serum as well as on other bodily samples. Finally, they also reported the possible role of microbiota derived EVs on asthma.  

The MS is very well conceived, structured and development and beside some minimal format revision it is, in my opinion, suitable for publication in IJMS. Here below my revisions:

1. Throughout the MS some double spaces seem present. Check it and correct.

2. In paragraph 3.2, lines 315-17, authors mentioned different immune cells, but they did not mention monocyte/macrophages. Nevertheless, shortly after (line 327 and following) they reported the role of alveolar macrophages. I suggest them to resentence lines 315-17 by adding also monocyte/macrophages.

3. In the paragraph corresponding to lanes 431-445 some font size mismatch is evident. Resize the font according to.

Author Response

Response to reviewer 2

We thank reviewer 2 for his/her appreciation of our manuscript. The response to the specific points raised are detailed below:

Reviewer 2 Point1. Throughout the MS some double spaces seem present. Check it and correct.

Response: thanks for picking up this inconsistency, we have edited the manuscript to a uniform space layout

Reviewer 2 Point2. In paragraph 3.2, lines 315-17, authors mentioned different immune cells, but they did not mention monocyte/macrophages. Nevertheless, shortly after (line 327 and following) they reported the role of alveolar macrophages. I suggest them to resentence lines 315-17 by adding also monocyte/macrophages.

Response: the reviewer is correct, we have added monocyte/macrophages to the paragraph.

Reviewer 2 Point3. In the paragraph corresponding to lanes 431-445 some font size mismatch is evident. Resize the font according to.

Response: thanks, we have edited the font size to a uniform one.